

# Global-scale evaluation of 23 precipitation datasets using gauge observations and hydrological modeling

Hylke E. Beck[1], Noemi Vergopolan[1], Ming Pan[1], Vincenzo Levizzani[2], Albert I.J.M. van Dijk[3], Graham Weedon[4], Luca Brocca[5], Florian Pappenberger[6], George J. Huffman[7], and Eric F. Wood[1]

[1]Department of Civil and Environmental Engineering, Princeton University, Princeton, New Jersey, USA
[2]National Research Council of Italy, Institute of Atmospheric Sciences and Climate (CNR-ISAC), Bologna, Italy
[3]Fenner School of Environment & Society, The Australian National University, Canberra, Australia
[4]Met Office, Joint Centre for Hydro-Meteorological Research, Wallingford, UK
[5]Research Institute for Geo-Hydrological Protection, National Research Council, Perugia, Italy
[6]European Centre for Medium-Range Weather Forecasts, Shinfield Park, Reading, UK
[7]Mesoscale Atmospheric Processes Laboratory, NASA Goddard Space Flight Center, Greenbelt, Maryland, USA

*Correspondence to:* Hylke E. Beck (hylkeb@princeton.edu)

**Abstract.** We undertook a comprehensive evaluation of 23 gridded (quasi-)global (sub-)daily precipitation ($P$) datasets for the period 2000–2016. Thirteen non-gauge-corrected $P$ datasets were evaluated using daily $P$ gauge observations from $76\,086$ gauges worldwide. Another ten gauge-corrected datasets were evaluated using hydrological modeling, by calibrating the conceptual model HBV against streamflow records for each of 9053 small to medium-sized ($< 50\,000\,\mathrm{km}^2$) catchments worldwide, and comparing the resulting performance. Marked differences in spatio-temporal patterns and accuracy were found among the datasets. Among the uncorrected $P$ datasets, the satellite- and reanalysis-based MSWEP-ng V1.2 and V2.0 datasets generally showed the best temporal correlations with the gauge observations, followed by the reanalyses (ERA-Interim, JRA-55, and NCEP-CFSR) and the the satellite- and reanalysis-based CHIRP V2.0 dataset, the estimates based primarily on passive microwave remote sensing of rainfall (CMORPH V1.0, GSMaP V5/6, and TMPA 3B42RT V7) or near-surface soil moisture (SM2RAIN-ASCAT), and finally, estimates based primarily on thermal infrared imagery (GridSat V1.0, PERSIANN, and PERSIANN-CCS). Two of the three reanalyses (ERA-Interim and JRA-55) unexpectedly obtained lower trend errors than the satellite datasets. Among the corrected $P$ datasets, the ones directly incorporating daily gauge data (CPC Unified and MSWEP V1.2 and V2.0) generally provided the best calibration scores, although the good performance of the fully gauge-based CPC Unified is unlikely to translate to sparsely or ungauged regions. Next best results were obtained with $P$ estimates directly incorporating temporally coarser gauge data (CHIRPS V2.0, GPCP-1DD V1.2, TMPA 3B42 V7, and WFDEI-CRU), which in turn outperformed those indirectly incorporating gauge data through other multi-source datasets (PERSIANN-CDR V1R1 and PGF). Our results highlight large differences in estimation accuracy, and hence, the importance of $P$ dataset selection in both research and operational applications. The good performance of MSWEP emphasizes that careful data merging can exploit the complementary strengths of gauge-, satellite- and reanalysis-based $P$ estimates.





## 1   Introduction

Precipitation ($P$) is arguably the most important driver of the hydrological cycle, but also one of the most challenging to estimate (Daly et al., 2008; Michaelides et al., 2009; Kidd and Levizzani, 2011; Tapiador et al., 2012). Over recent decades, several gridded $P$ datasets have been developed that are suitable for large-scale hydrological applications (for overviews, see Table 1, Beck et al., 2017c, http://ipwg.isac.cnr.it, and http://reanalyses.org). The datasets differ in terms of design objective (temporal homogeneity, instantaneous accuracy, or both), data sources (radar, gauge, satellite, analysis, or reanalysis, or combinations thereof), spatial resolution (from 0.05° to 2.5°), spatial coverage (from continental to fully global), published temporal resolution (from 30 minutes to monthly), temporal span (from ∼1 to 115 years), and latency (from ∼3 hours to several years).

A plethora of studies addressed the important task of evaluating these $P$ datasets to understand their respective advantages and limitations (see reviews by Gebremichael, 2010, and Maggioni et al., 2016). Most studies assessed accuracy using independent gauge observations (e.g., Hirpa et al., 2010; Buarque et al., 2011; Bumke et al., 2016; Alijanian et al., 2017) or gauge-adjusted radar fields (e.g., AghaKouchak et al., 2011; Islam et al., 2012), while others merely compared their spatio-temporal patterns (e.g., Kidd et al., 2013). Yet others quantified the performance of different $P$ datasets using hydrological modeling, by comparing simulated and observed values of river discharge ($Q$; e.g., Collischonn et al., 2008; Behrangi et al., 2011; Bitew et al., 2012; Falck et al., 2015) or soil moisture (e.g., Pan et al., 2010; Albergel et al., 2013; Martens et al., 2017). More recently, Massari et al. (2017) assessed the performance of different $P$ datasets using triple collocation. Marked differences in spatio-temporal $P$ patterns and accuracy have been found among the datasets, even among those employing the same data sources. This highlights the critical importance of dataset choice for research and operational applications alike.

Previous evaluation studies used a wide variety of evaluation approaches and performance metrics (Ebert, 2007; Gebremichael, 2010; Loew et al., 2017). However, many studies considered only a single $P$ dataset (e.g., Scheel et al., 2011; Nair and Indu, 2017) or disregarded (re)analysis-based $P$ datasets (e.g., Moazami et al., 2013; Mei et al., 2014; Zambrano-Bigiarini et al., 2017), despite their demonstrated superior performance in cold climates (Ebert et al., 2007; Beck et al., 2017c; Massari et al., 2017). In addition, some studies re-used gauge observations already incorporated in some of the $P$ datasets to determine their accuracy (e.g., Chen et al., 2013; Ashouri et al., 2016; Zambrano-Bigiarini et al., 2017), precluding independent validation. Furthermore, to our knowledge, so far no study has accounted for differences in the exact UTC boundary of the 24-hour accumulation period of daily gauge reports when evaluating $P$ datasets, potentially confounding the results. Moreover, studies employing hydrological modeling generally used $Q$ observations from a small number of catchments (e.g., Bitew et al., 2012, and Tang et al., 2016, both used only one) and did not attempt to recalibrate the hydrological model for each $P$ dataset individually (e.g., Su et al., 2008; Li et al., 2013), leading to combined rainfall and model uncertainty that is not easily interpreted. Finally, many have a regional (sub-continental) focus (Maggioni et al., 2016), and therefore it is not clear to what extent the results can be generalized.

Nevertheless, there have also been several (quasi-)global $P$ dataset evaluation studies that produced general insights (e.g., Adler et al., 2001; Fekete et al., 2004; Voisin et al., 2008; Bosilovich et al., 2008; Tian and Peters-Lidard, 2010; Lorenz and Kunstmann, 2012; Yong et al., 2015; Herold et al., 2015; Gehne et al., 2016; Massari et al., 2017). These studies revealed that





satellites (resp. reanalyses) exhibit superior performance at low (high) latitudes dominated by intense, localized convective (persistent, large-scale stratiform) $P$ systems. However, none of these studies took advantage of the vast amount of $P$ gauge observations contained in the freely available GHCN-D (Menne et al., 2012) and GSOD (https://data.noaa.gov) databases. Among the only two studies employing hydrological modeling, Fekete et al. (2004) performed monthly simulations and did

not compare the results against observed $Q$, while Voisin et al. (2008) used monthly observed $Q$ data from only nine very large catchments ($> 290\,000$ km$^2$). Moreover, several promising recently released or revised $P$ datasets, such as CHIRPS V2.0, MSWEP V2.0, and PERSIANN-CDR V1R1 (see Table 1), have not been thoroughly evaluated yet at a (quasi-)global scale.

Our objective was to undertake the most comprehensive global-scale $P$ dataset evaluation to date. We evaluated thirteen non-gauge-corrected $P$ datasets using daily $P$ gauge observations from 76 086 gauges worldwide. Another ten gauge-corrected

$P$ datasets were evaluated using hydrological modeling for 9053 catchments ($< 50\,000$ km$^2$) worldwide, by calibrating a hydrological model. The expectation was that such a large number of $P$ datasets and large number of observations should lead to more generally valid conclusions, and allows us to explicitly compare the performance among climate types and regions (Andréassian et al., 2007; Gupta et al., 2014).

## 2 Data and methods

### 2.1 $P$ datasets

Table 1 presents the 23 gridded $P$ datasets included in the evaluation. The datasets were classified as either uncorrected, meaning that their temporal dynamics depend entirely on satellite and/or reanalysis data, or gauge-corrected, meaning that their temporal dynamics depend at least partly on gauge data (hence precluding an independent evaluation using $P$ gauge observations). For clarity and reproducibility, we report dataset version numbers throughout the study for the datasets for

which this information was available. We only included datasets with a temporal span of $> 8$ years.

### 2.2 Performance evaluation using gauge observations

The performance of the thirteen uncorrected $P$ datasets (see Table 1) was evaluated using daily gauge observations from across the globe. Our collection of gauge observations was compiled from the Global Historical Climatology Network-Daily (GHCN-D) database (Menne et al., 2012), the Global Summary of the Day (GSOD) database (https://data.noaa.gov), the Latin American

Climate Assessment & Dataset (LACA&D) database (http://lacad.ciifen-int.org), the Chile Climate Data Library (http://www. climatedatalibrary.cl), and national databases for Mexico, Brazil, Peru, and Iran. To discard erroneous observations, each gauge record was subjected to several quality checks as described in Beck et al. (2017a). Only gauges with $> 365$ days of valid data (not necessarily consecutive) during 2000–2016 were retained. To minimize temporal mismatches in gauge and gridded $P$ time series, we used the gauge reporting times from Beck et al. (2017a) to shift the records of gauges with reporting times

$> +12$ hours UTC backward by one day, and the records of gauges with reporting times $< -12$ hours UTC forward by one day. In total 76 086 gauges had sufficient quality-controlled data for the evaluation.

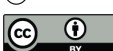

**Table 1.** Overview of the 23 (quasi-)global (sub-)daily gridded *P* datasets evaluated in this study. Abbreviations in the data source(s) column defined as: G, gauge; S, satellite; and R, reanalysis. The acronym NRT in the temporal coverage column stands for Near Real-Time.

| Short name | Full name and details | Data source(s) | Spatial resolution | Spatial coverage | Temporal resolution | Temporal coverage | Reference |
|---|---|---|---|---|---|---|---|
| **Non-gauge-corrected datasets** | | | | | | | |
| CHIRP V2.0 | Climate Hazards group Infrared Precipitation (CHIRP) V2.0 (http://chg.ucsb.edu/data/chirps/) | S, R | 0.05° | Land, < 50° | Daily | 1981–NRT[2] | Funk et al. (2015a) |
| CMORPH V1.0 | CPC MORPHing technique (CMORPH) V1 (www.cpc.ncep.noaa.gov) | S | 0.07° | < 60° | 30 min. | 1998–NRT[1] | Joyce et al. (2004) |
| ERA-Interim | European Centre for Medium-range Weather Forecasts ReAnalysis Interim (ERA-Interim; https://www.ecmwf.int/en/research/climate-reanalysis/era-interim) | R | ~0.75° | Global | 3 hourly | 1979–2017[3] | Dee et al. (2011) |
| GSMaP V5/6 | Global Satellite Mapping of Precipitation (GSMaP) Moving Vector with Kalman (MVK) standard V5 and V6 (http://sharaku.eorc.jaxa.jp/GSMaP/) | S | 0.1° | < 60° | Hourly | 2000–NRT[1] | Ushio et al. (2009) |
| GridSat V1.0 | *P* derived from the Gridded Satellite (GridSat) B1 thermal infrared archive v02r01 (Knapp et al., 2011; https://www.ncdc.noaa.gov/gridsat/) | S | 0.1° | < 50° | 3 hourly | 1983–2016 | Beck et al. (2017a) |
| JRA-55 | Japanese 55-year ReAnalysis (JRA-55; jra.kishou.go.jp/JRA-55) | R | ~0.56° | Global | 3 hourly | 1959–NRT[2] | Kobayashi et al. (2015) |
| MSWEP-ng V1.2 | Multi-Source Weighted-Ensemble Precipitation (MSWEP) no-gauge (ng) V1.2 (www.gloh2o.org) | S, R | 0.25° | Global | 3 hourly | 1979–2015 | Beck et al. (2017c) |
| MSWEP-ng V2.0 | Multi-Source Weighted-Ensemble Precipitation (MSWEP) no-gauge (ng) V2.0 (www.gloh2o.org) | S, R | 0.1° | Global | 3 hourly | 1979–NRT[1] | Beck et al. (2017a) |
| NCEP-CFSR | National Centers for Environmental Prediction (NCEP) Climate Forecast System Reanalysis (CFSR; http://cfs.ncep.noaa.gov/cfsr/) | R | ~0.31° | Global | Hourly | 1979–2010 | Saha et al. (2010) |
| PERSIANN | Precipitation Estimation from Remotely Sensed Information using Artificial Neural Networks (PERSIANN; http://chrs.web.uci.edu) | S | 0.25° | < 60° | Hourly | 2000–NRT[1] | Sorooshian et al. (2000) |
| PERSIANN-CCS | Precipitation Estimation from Remotely Sensed Information using Artificial Neural Networks (PERSIANN) Cloud Classification System (CCS; http://chrs.web.uci.edu) | S | 0.04° | < 60° | Hourly | 2003–NRT[1] | Hong et al. (2004) |
| SM2RAIN-ASCAT | *P* inferred from Advanced Scatterometer (ASCAT) satellite near-surface soil moisture (http://hydrology.irpi.cnr.it) | S | 0.5° | Land | Daily | 2007–2015 | Brocca et al. (2014) |
| TMPA 3B42RT V7 | TRMM Multi-satellite Precipitation Analysis (TMPA) 3B42RT V7 (https://mirador.gsfc.nasa.gov) | S | 0.25° | < 50° | 3 hourly | 2000–NRT[1] | Huffman et al. (2007) |
| **Gauge-corrected datasets** | | | | | | | |
| CHIRPS V2.0 | Climate Hazards group Infrared Precipitation with Stations (CHIRPS) V2.0 (http://chg.ucsb.edu/data/chirps/) | G, S, R | 0.05° | Land, < 50° | Daily | 1981–NRT[2] | Funk et al. (2015a) |
| CMORPH-CRT V1.0 | CPC MORPHing technique (CMORPH) bias corrected (CRT) V1.0 (www.cpc.ncep.noaa.gov) | G, S | 0.07° | < 60° | 30 min. | 1998–2015 | Not available |
| CPC Unified | Climate Prediction Center (CPC) Unified V1.0 and RT (https://www.esrl.noaa.gov/psd/data/gridded/) | G | 0.5° | Land | Daily | 1979–NRT[2] | Chen et al. (2008) |
| GPCP-1DD V1.2 | Global Precipitation Climatology Project (GPCP) 1-Degree Daily (1DD) Combination V1.2 (https://precip.gsfc.nasa.gov) | G, S | 1° | Global | Daily | 1996–2015 | Huffman et al. (2001) |
| MSWEP V1.2 | Multi-Source Weighted-Ensemble Precipitation (MSWEP) V1.2 (www.gloh2o.org) | G, S, R | 0.25° | Global | 3 hourly | 1979–2015 | Beck et al. (2017c) |
| MSWEP V2.0 | Multi-Source Weighted-Ensemble Precipitation (MSWEP) V2.0 (www.gloh2o.org) | G, S, R | 0.1° | Global | 3 hourly | 1979–NRT[1] | Beck et al. (2017a) |
| PERSIANN-CDR V1R1 | Precipitation Estimation from Remotely Sensed Information using Artificial Neural Networks (PERSIANN) Climate Data Record (CDR) V1R1 (http://chrs.web.uci.edu) | G, S | 0.25° | < 60° | 6 hourly | 1983–2016 | Ashouri et al. (2015) |
| PGF | Princeton Global meteorological Forcing dataset (http://hydrology.princeton.edu) | G, R | 0.25° | Global | 3 hourly | 1948–2012 | Sheffield et al. (2006) |
| TMPA 3B42 V7 | TRMM Multi-satellite Precipitation Analysis (TMPA) 3B42 V7 (https://mirador.gsfc.nasa.gov/) | G, S | 0.25° | < 50° | 3 hourly | 2000–2017[3] | Huffman et al. (2007) |
| WFDEI-CRU | WATCH Forcing Data ERA-Interim (WFDEI; www.eu-watch.org) | G, R | 0.5° | Global | 3 hourly | 1979–2015 | Weedon et al. (2014) |

[1] Available until the present with a delay of several hours.
[2] Available until the present with a delay of several days.
[3] Available until the present with a delay of several months.





We considered the following five performance metrics to evaluate the $P$ datasets in terms of temporal dynamics: (i) Pearson linear correlation coefficient ($R$) calculated for 3-day means ($R_{3\,day}$); (ii) $R$ calculated for monthly means ($R_{monthly}$); (iii) $R$ calculated for 6-month Standardized Precipitation Index values ($R_{SPI-6}$; Hayes et al., 1999); (iv) Mean Absolute Error (MAE; mm month$^{-1}$) for monthly means; and (v) the trend error (the difference between gauge- and dataset-based linear regression slopes calculated from annual anomalies; % yr$^{-1}$). We opted for MAE instead of the more widely used Root Mean Square Error (RMSE) because the errors are unlikely to follow a normal distribution (Chai and Draxler, 2014; Willmott et al., 2017). We used 3-day rather than daily means for $R_{3\,day}$ to minimize the impact of any residual mismatches in the UTC boundary of the 24-hour accumulation period between the gauges and datasets. The $R_{3\,day}$ metric was only calculated if $\geq 60$ 3-day contemporaneous gauge and dataset values were available, while the $R_{monthly}$, $R_{SPI-6}$, and MAE metrics were only calculated if $\geq 12$ monthly contemporaneous gauge and dataset values were available.

To evaluate the $P$ datasets in terms of long-term mean climate indices, we considered the following four metrics: (i) long-term relative bias, defined as $\left[\overline{s}-\overline{o}\right]/\left[\overline{s}+\overline{o}\right]$, where $\overline{s}$ and $\overline{o}$ represent the dataset- and gauge-based long-term means, respectively; (ii) annual number of dry days error (using a 0.5-mm-d$^{-1}$ threshold to identify dry days, similar to Akinremi et al., 1999, Haylock et al., 2008, and Driouech et al., 2009); and (iii) 99th and 99.9th percentile daily $P$ error (mm d$^{-1}$). The bias and trend error metrics were only calculated if $> 5$ years of daily contemporaneous gauge and dataset values were available.

## 2.3 Performance evaluation using hydrological modeling

The performance of the ten gauge-corrected $P$ datasets (see Table 1) was evaluated using hydrological modeling for 9053 catchments. Our collection of $Q$ observations was compiled from the same three sources as Beck et al. (2015), viz.: (i) the US Geological Survey (USGS) Geospatial Attributes of Gages for Evaluating Streamflow (GAGES)-II database (Falcone et al., 2010); (ii) the Global Runoff Data Centre (GRDC; http://www.bafg.de/GRDC/); and (iii) the Australian Peel et al. (2000) database. We only used catchments $< 50\,000$ km$^2$ with a $Q$ record length $> 365$ days (not necessarily consecutive) during 2000–2012 (the common temporal coverage of the $P$ datasets), resulting in 9053 catchments that were suitable for the evaluation (5th, 50th, and 95th percentile catchment size of 9, 633, and 18468 km$^2$, respectively).

For each catchment, the conceptual hydrological model HBV (Bergström, 1992; Seibert and Vis, 2012) was calibrated in a lumped fashion against $Q$ observations using daily $P$ time series from each of the datasets to force the model. The model was selected because of its agility, computational efficiency, and widespread successful application (e.g., Te Linde et al., 2008; Deelstra et al., 2010; Plesca et al., 2012; Beck et al., 2013; Valéry et al., 2014; Vetter et al., 2015; Beck et al., 2017b). For the calibration, we employed the $(\mu + \lambda)$ evolutionary algorithm (Ashlock, 2010; Fortin et al., 2012) with the population size ($\mu$) set to 20, the recombination pool size ($\lambda$) set to 40, and the number of generations set to 12 (amounting to 480 model runs per catchment per $P$ dataset and approximately 40 million model runs in total). As objective function we used the common Nash and Sutcliffe (1970) Efficiency (NSE) computed between 3-day mean simulated and observed $Q$ time series. We used 3-day rather than daily mean $Q$ time series for the NSE calculation to reduce the impact of temporal mismatches in simulated and observed $Q$ peaks. A higher calibration NSE generally implies that the $P$ dataset in question is more consistent with the $Q$ observations and potential evaporation ($E_p$) estimates and thus that the $P$ dataset is more accurate. See Beck et al. (2016) and



Beck et al. (2017c) for more details on the hydrological model, calibration algorithm, model parameter ranges, $Q$ observations, $E_p$ forcing, and $T_a$ forcing. We recognize that using data from different sources may bias results as the water balances are unlikely to be closed.

## 3 Results and Discussion

### 3.1 Performance for temporal dynamics

The temporal dynamics of the thirteen uncorrected $P$ datasets were evaluated using daily $P$ observations from 76 086 gauges around the globe. Table 2 presents summary statistics separately for the gauges located at latitudes $< 40°$ for all datasets, and for the gauges located at latitudes $\geq 40°$ only for the datasets covering the entire terrestrial surface (i.e., MSWEP-ng V1.2 and V2.0, and the reanalyses). In terms of temporal correlations ($R_{3\,\text{day}}$, $R_{\text{monthly}}$, and $R_{\text{SPI-6}}$), the satellite- and reanalysis-based MSWEP-ng datasets performed overall slightly better than the reanalyses (ERA-Interim, JRA-55, and NCEP-CFSR) and the satellite- and reanalysis-based CHIRP V2.0 dataset, which in turn performed slightly better than the satellite datasets based primarily on passive microwave retrievals (CMORPH V1.0, GSMaP V5/6, and TMPA 3B42RT V7) and near-surface soil moisture (SM2RAIN-ASCAT), which in turn performed slightly better than the satellite datasets based primarily on thermal infrared imagery (GridSat V1.0, PERSIANN, and PERSIANN-CCS). The high correlations obtained using both versions of MSWEP-ng underscore the effectiveness of merging multiple satellite and reanalysis datasets (Beck et al., 2017c, a). In agreement with our results, Stillman et al. (2016) found reanalyses to outperform infrared- and passive microwave-based satellite datasets in Arizona. Contrary to expectation, PERSIANN-CCS attained lower median correlations than both GridSat V1.0 and PERSIANN, despite using a more sophisticated algorithm and higher spatial resolution (Hong et al., 2004). The better performance of the microwave-based datasets compared to infrared-based ones is in line with previous evaluations (e.g., Hirpa et al., 2010; Peña Arancibia et al., 2013; Cattani et al., 2016) and attributed to the indirect relationship between cloud-top infrared brightness temperatures and surface rainfall (Stephens and Kummerow, 2007). SM2RAIN-ASCAT was found to perform similar to TMPA 3B42RT V7, in agreement with Brocca et al. (2014), suggesting that soil moisture-based approaches provide a promising additional source of rainfall data.

Figure 1 presents global $R_{3\,\text{day}}$ maps for a selection of eight $P$ datasets, permitting a geographical interpretation of the results (see the Supplementary information for global maps of the other performance metrics). All datasets performed relatively poorly ($R_{3\,\text{day}} < 0.5$) in arid and tropical regions, due to the often highly localized and shortlived nature of the convective rainfall that dominates. Sub-cloud evaporation of falling rain potentially constitutes an additional confounding factor in arid regions (Dinku et al., 2016). Conversely, all datasets performed relatively well ($R_{3\,\text{day}} \geq 0.5$) in moist mid-latitude regions with mild winters (e.g., the southeastern US, eastern South America, and eastern China). In accordance with several previous global evaluations (e.g., Barrett et al., 1994; Xie and Arkin, 1997; Adler et al., 2001; Ebert et al., 2007; Massari et al., 2017), the reanalyses exhibited lower skill levels than the microwave- and infrared-based satellite datasets in the tropics, whereas the opposite is true for colder regions (latitudes $> 40°$). Africa showed the lowest $R_{3\,\text{day}}$ values overall, which is concerning because it is also the most poorly gauged continent (Thiemig et al., 2012; Sylla et al., 2013; Kidd et al., 2017).



MSWEP V2.0 obtained substantially lower mean annual $P$ trend errors than the other $P$ datasets (Table 2 and Supplementary information Figure S5). Two of the three reanalyses (ERA-Interim and JRA-55) provided more reliable trends than the satellite datasets, contrary to the common assumption that reanalyses tend to contain temporal discontinuities due to changes in the assimilated observations (Bengtsson et al., 2004; Lorenz and Kunstmann, 2012; Kang and Ahn, 2015). However, our evaluation covers a relatively short period (2000–2016) during which the assimilated observations may not have changed much. Among the satellite datasets, SM2RAIN-ASCAT provided the least accurate $P$ trends, probably due to the use of two ASCAT sensors after 2013 (on-board MetOp-A and MetOp-B) which artificially increased rainfall amounts obtained using SM2RAIN (separate calibrations for 2007–2012 and 2013–2015 are necessary but yet to be performed). Among the reanalyses, NCEP-CFSR performed worst. Following previous authors (Saha et al., 2010; Wang et al., 2013), we speculate that this may be attributable to the six parallel-run streams of analysis covering different periods, which have been combined to generate the final dataset. The relatively small mean annual $P$ trend errors obtained for the different datasets (ranging from 1.53–3.56 % yr$^{-1}$) provide some confidence in the ability to infer significant trends from the various datasets. However, trends for variables measured over shorter temporal scales (e.g., annual maxima or percentiles) are likely to be subject to much greater uncertainty.

### 3.2 Performance for climate indices

The performance of the thirteen uncorrected $P$ datasets in terms of several long-term climate indices is summarized in Table 2, listing summary statistics for $P$ gauges at latitudes $< 40°$ and $\geq 40°$ (for the five datasets covering the entire terrestrial surface), respectively. In terms of bias, the reanalyses performed better overall than the satellite datasets (Table 2). Although CHIRP V2.0, GridSat V1.0, and MSWEP-ng V1.2 and V2.0 obtained the best bias scores, these datasets use the gauge-based CHPclim (Funk et al., 2015b) or WorldClim (Fick and Hijmans, 2017) datasets to determine their long-term mean. The spread in the range of bias scores among the datasets was generally greatest over topographically complex regions (notably the Rocky, Andes, and Hindu-Kush Mountains), and in arid regions (notably the Sahara, Arabian, and Gobi deserts; Supplementary information Figure S6), demonstrating the particular difficulty of estimating $P$ in these regions (Fekete et al., 2004; Hirpa et al., 2010; Xu et al., 2017; Kim et al., 2017). All fully global datasets exhibited positive biases at high northern latitudes, probably because the $P$ gauge data used for evaluation were not corrected for wind-induced under-catch (Groisman and Legates, 1994; Rasmussen et al., 2012; Kauffeldt et al., 2013).

In terms of the annual number of dry days, the datasets exhibited a particularly large spread in performance, with MSWEP-ng V2.0 outperforming the other datasets by a substantial margin (Table 2 and Supplementary information Figure S7). The dramatic improvement in MSWEP-ng V2.0 compared to V1.2 is mainly attributable to the cumulative distribution function corrections introduced in V2, which eliminate the drizzle caused by averaging multiple data sources (Beck et al., 2017a). The infrared- and microwave-based satellite datasets also performed reasonably well, although the $P$ frequency was generally overestimated at low and mid latitudes and underestimated at high latitudes, reflecting the difficulty of detecting $P$ signals at high latitudes (Ferraro et al., 1998; Ebert et al., 2007; Kidd and Levizzani, 2011; Kidd et al., 2012; Laviola et al., 2013). Conversely, the reanalyses consistently underestimated the number of dry days across the globe, due to the presence of spurious drizzle caused by deficiencies in the representation and/or parameterization of the physical processes governing $P$ generation





(Zolina et al., 2004; Lopez, 2007; Sun et al., 2006; Skok et al., 2015). SM2RAIN-ASCAT also consistently underestimated the number of dry days due to the presence of spurious drizzle, in this case due to the relatively noisy soil moisture retrievals (Crow et al., 2011; Brocca et al., 2014) and the use of the already fairly wet ERA-Interim dataset for the algorithm calibration. CHIRP V2.0 also exhibited too few dry days, which is attributed to the use of TMPA 3B42 for establishing the regression equations linking infrared-based cold-cloud duration values to 5-day mean $P$ (Funk et al., 2015a). Forcing a hydrological model with $P$ data overestimating the frequency of low-intensity rainfall events is likely to result in overestimated evaporation and underestimated runoff, particularly in regions with high soil or canopy water storage capacities.

The 99th and 99.9th percentile daily $P$ errors measure the error in the magnitude of storms with return periods of 100 days and 2.7 years, respectively (Table 2, and Supplementary information Figures S8 and S9, respectively). MSWEP-ng V2.0 performed best in this respect, whereas CHIRP V2.0, the reanalyses, MSWEP-ng V1.2, and particularly SM2RAIN-ASCAT consistently underestimated the 99th and 99.9th percentile storm magnitudes. However, some degree of underestimation would be expected, given the spatial scale mismatch between gauge observations and grid-cell averages (see, e.g., Maraun, 2013), particularly for $P$ datasets with a coarse spatial resolution (see Table 1). Nevertheless, for the reanalyses the underestimation is probably primarily attributable to the aforementioned model uncertainties. For MSWEP-ng V1.2, it is due to the attenuating effect of merging multiple data sources (Beck et al., 2017c). For SM2RAIN-ASCAT, the strong underestimation of storm magnitudes may at least partly be due to signal loss induced by soil saturation (Brocca et al., 2014). Among the microwave- and infrared-based satellite datasets, PERSIANN-CCS showed the greatest spatial variability in storm magnitude bias. The generally strong differences in spatial performance patterns among datasets highlight the difficulty of generalizing the findings of regional (sub-continental) evaluation studies.

## 3.3 Performance evaluation using hydrological modeling

The performance of the ten gauge-corrected $P$ datasets (see Table 1) was evaluated using hydrological modeling for 9053 catchments around the globe. Table 3 presents median calibration NSE scores obtained using the different $P$ datasets for different climate zones. The overall performance ranking of the datasets from best to worst (% of catchments in which the dataset performed best between parentheses) is MSWEP V2.0 (44.9 %), MSWEP V1.2 (21.3 %), CPC Unified (15.7 %), WFDEI-CRU (4.9 %), TMPA 3B42 V7 (3.3 %), CMORPH-CRT V1.0 (2.6 %), CHIRPS V2.0 (2.5 %), PERSIANN-CDR V1R1 (2.1 %), GPCP-1DD V1.2 (1.6 %), and PGF (1.1 %). Thus, the datasets directly incorporating daily gauge data (CPC Unified, and MSWEP V1.2 and V2.0) overall outperformed the ones directly incorporating 5-day (CHIRPS V2.0) or monthly (GPCP-1DD V1.2, TMPA 3B42 V7, and WFDEI-CRU) gauge data, which in turn outperformed PERSIANN-CDR V1R1 and PGF. Rather than using gauge observations directly for corrections, PERSIANN-CDR V1R1 is adjusted to match the satellite- and gauge-based GPCP dataset (monthly temporal and 2.5° spatial resolution), whereas PGF employs the gauge-satellite-based GPCP-1DD dataset only to spatially disaggregate the daily estimates. An additional reason for the relatively poor performance of GPCP-1DD V1.2 may be its coarse 1° spatial resolution. For PGF, additional reasons are the use of a correction methodology that compromises the daily variability (Sheffield et al., 2004) and the fact that it is based on a first generation reanalysis (NCEP/NCAR Reanalysis 1; Kistler et al., 2001; Sheffield et al., 2006). It is noted that some of the datasets, such



**Table 2.** Median values of the performance metrics for the uncorrected $P$ datasets based on daily $P$ observations from 76 086 gauges around the globe. Statistics were not shown for the satellite-based $P$ datasets for the group of gauges located at latitudes $\geq 40°$. For all performance metrics, with the exception of $R_{3\,\mathrm{day}}$, $R_{\mathrm{monthly}}$, and $R_{\mathrm{SPI}\text{-}6}$, a lower value represents better performance. Values in bold represent the best score for each metric. See the Supplementary information for global maps with scores for the performance metrics for a selection of eight $P$ datasets.

| | CHIRP V2.0 | CMORPH V1.0 | ERA-Interim V1.0 | GridSat V1.0 | GSMaP V5/6 | JRA-55 | MSWEP-ng V1.2 | MSWEP-ng V2.0 | NCEP-CFSR | PERS. | PERS.-CCS | SM2RAIN-ASCAT | 3B42RT V7 |
|---|---|---|---|---|---|---|---|---|---|---|---|---|---|
| **Gauges located at latitudes $< 40°$ ($n = 51\,271$)** | | | | | | | | | | | | | |
| $R_{3\,\mathrm{day}}$ (—) | 0.55 | 0.53 | 0.59 | 0.44 | 0.54 | 0.56 | **0.67** | 0.64 | 0.57 | 0.47 | 0.42 | 0.52 | 0.52 |
| $R_{\mathrm{monthly}}$ (—) | 0.74 | 0.69 | 0.75 | 0.60 | 0.69 | 0.75 | **0.82** | 0.81 | 0.75 | 0.62 | 0.59 | 0.68 | 0.69 |
| $R_{\mathrm{SPI}\text{-}6}$ (—) | 0.71 | 0.65 | 0.74 | 0.60 | 0.67 | 0.72 | **0.81** | 0.80 | 0.72 | 0.58 | 0.56 | 0.68 | 0.66 |
| MAE (mm month$^{-1}$) | 30.54 | 37.81 | 31.41 | 43.79 | 36.10 | 32.87 | **26.96** | 27.99 | 32.32 | 42.53 | 45.51 | 36.67 | 37.46 |
| Trend error (% yr$^{-1}$) | 1.87 | 2.23 | 1.97 | 2.34 | 3.34 | 1.91 | 1.61 | **1.53** | 3.56 | 2.68 | 2.46 | 3.39 | 2.14 |
| Bias (—) | **0.06** | 0.14 | 0.11 | 0.07 | 0.13 | 0.11 | **0.06** | **0.06** | 0.10 | 0.17 | 0.17 | 0.14 | 0.11 |
| Annual dry days error (days) | 73.85 | 15.77 | 47.49 | 21.55 | 20.90 | 43.22 | 65.06 | **10.46** | 37.95 | 27.65 | 28.49 | 112.36 | 17.63 |
| 99th percentile error (mm d$^{-1}$) | 13.02 | 7.27 | 13.73 | 4.71 | 7.54 | 8.71 | 11.01 | **4.59** | 7.37 | 9.69 | 8.97 | 26.00 | 6.18 |
| 99.9th percentile error (mm d$^{-1}$) | 34.65 | 17.21 | 27.82 | 15.87 | 18.54 | 24.66 | 29.30 | **14.90** | 16.09 | 21.64 | 20.24 | 63.38 | 15.83 |
| **Gauges located at latitudes $\geq 40°$ ($n = 24\,815$)** | | | | | | | | | | | | | |
| $R_{3\,\mathrm{day}}$ (—) | — | — | 0.68 | — | — | 0.67 | **0.74** | 0.72 | 0.66 | — | — | — | — |
| $R_{\mathrm{monthly}}$ (—) | — | — | 0.78 | — | — | 0.79 | **0.84** | 0.83 | 0.73 | — | — | — | — |
| $R_{\mathrm{SPI}\text{-}6}$ (—) | — | — | 0.77 | — | — | 0.78 | **0.82** | **0.82** | 0.73 | — | — | — | — |
| MAE (mm month$^{-1}$) | — | — | 21.56 | — | — | 24.17 | **19.25** | 19.70 | 26.60 | — | — | — | — |
| Trend error (% yr$^{-1}$) | — | — | 1.41 | — | — | 1.35 | 1.27 | **1.20** | 2.20 | — | — | — | — |
| Bias (—) | — | — | 0.09 | — | — | 0.10 | **0.05** | **0.05** | 0.11 | — | — | — | — |
| Annual dry days error (days) | — | — | 45.85 | — | — | 41.93 | 58.14 | **7.79** | 55.79 | — | — | — | — |
| 99th percentile error (mm d$^{-1}$) | — | — | 6.26 | — | — | 3.80 | 6.10 | **3.06** | 3.59 | — | — | — | — |
| 99.9th percentile error (mm d$^{-1}$) | — | — | 15.95 | — | — | 12.52 | 16.80 | **9.22** | 9.83 | — | — | — | — |





**Figure 1.** For a selection of the evaluated uncorrected $P$ datasets, temporal correlations between 3-day mean gauge- and dataset-based $P$ time series ($R_{3\,day}$). Each data point represents a gauge. See the Supplementary information for global maps of the other performance metrics.



as CHIRPS V2.0 and PERSIANN-CDR V1R1, have not been specifically designed to provide the best instantaneous accuracy, but rather to achieve the most temporally homogeneous record possible. Furthermore, the good performance of the exclusively gauge-based CPC Unified is unlikely to generalize to regions with sparse rain gauge networks.

Figure 2 presents global maps with calibration NSE values obtained for a selection of the best performing $P$ datasets,
while Figure 3 shows which of these $P$ datasets obtained the highest calibration NSE for each catchment. All $P$ datasets provided low calibration NSE scores ($< 0.3$) over the US Great Plains, consistent with several previous studies using different hydrological models and forcing datasets (e.g., Newman et al., 2015; Bock et al., 2015; Essou et al., 2016). It reflects the spatio-temporally highly intermittent rainfall regime combined with a strongly non-linear rainfall-runoff response (Pilgrim et al., 1988). Low calibration scores were also found in northern Alaska, presumably due to $P$ underestimation (Kauffeldt
et al., 2013); in Namibia and Zambia, probably partly due to the importance of convective rainfall and partly due to the $Q$ data quality (Li et al., 2013); and in Hawaii, we suspect due to flow overestimations caused by erroneous rating curves, as visual inspection of the records revealed the presence of drift errors. In North America, Europe, Japan, Australia, New Zealand, and southern and western Brazil MSWEP V2.0 generally exhibited the best performance, whereas in Central America, and in central and eastern Brazil CHIRPS V2.0 tended to perform best. No obviously best estimate could be identified for Africa,
emphasizing the challenge of hydrological modeling in Africa (Sylla et al., 2013; Beck et al., 2017b). In summary, there are some $P$ datasets that consistently outperform others regionally, but there is not one that performs best everywhere (Barrett et al., 1994).

The good performance obtained for CPC Unified, CHIRPS V2.0, and MSWEP V1.2 and V2.0 underscores the importance of using sub-monthly gauge observations to improve $Q$ simulations. Few $P$ datasets currently incorporate sub-monthly gauge
data, possibly because of the better global-scale availability of monthly gauge data, the lack of reliable information on the 24-hour accumulation time for the large majority of gauges across the globe, and the difficulty of applying daily rather than monthly gauge corrections (Vila et al., 2009). However, a wealth of daily gauge data is currently freely available (Menne et al., 2012; Funk et al., 2015a), and sub-daily satellite and reanalysis $P$ estimates provide an efficient and consistent means to infer the most probable UTC boundary of the 24-hour accumulation period for any gauge with observations during the satellite era
(1979–present; Beck et al., 2017a).

Most previous studies using hydrological modeling to evaluate the accuracy of $P$ datasets have a regional or sub-continental focus, used $Q$ observations from a relatively small number of catchments, considered only a few $P$ datasets, did not consider reanalysis-based $P$ datasets, or did not re-calibrate the hydrological model for each $P$ dataset (e.g., Voisin et al., 2008; Su et al., 2008; Bitew et al., 2012; Tang et al., 2016). Here, we used 9053 catchments covering all climate zones and latitudes,
considered a diverse range of $P$ datasets, and re-calibrated the model for each $P$ dataset, to maximize the generalizability of our findings. Nevertheless, our catchments are predominantly located in regions with dense $P$ gauge networks (i.e., the conterminous US, Europe, and parts of Australia). Therefore, our results may not unequivocally generalize to regions with sparse $P$ gauge networks. Use of another calibration objective function, hydrological model, or $T_a$ or $E_p$ forcing may have led to slightly different results, although we consider it unlikely to change the overall performance ranking of the $P$ datasets.
Finally, a poor score for a particular $P$ dataset may also simply reflect a systematic bias that could be easily corrected.

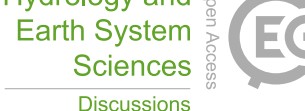



**Figure 2.** Calibration NSE scores obtained using $P$ time series from (a) CHIRPS V2.0, (b) CMORPH-CRT V1.0, (c) CPC Unified, (d) MSWEP V1.2, (e) MSWEP V2.0, (f) PERSIANN-CDR V1R1, (g) TMPA 3B42 V7, and (h) WFDEI-CRU. Each data point represents a catchment centroid. Only the eight best performing $P$ datasets are shown.




**Table 3.** Median calibration NSE scores for the gauge-corrected $P$ datasets obtained using HBV. Only the catchments with calibration NSE values for all $P$ datasets are considered. Thus, catchments at latitudes $> 50°$ have been excluded. Values in bold represent the highest score on each row.

| Köppen-Geiger climate zone | Number of catchments | CHIRPS V2.0 | CMORPH-CRT V1.0 | CPC Unified | GPCP-1DD V1.2 | MSWEP V1.2 | MSWEP V2.0 | PERSIANN-CDR V1R1 | PGF | TMPA 3B42 V7 | WFDEI-CRU |
|---|---|---|---|---|---|---|---|---|---|---|---|
| All | 8220 | 0.45 | 0.17 | 0.54 | 0.27 | 0.58 | **0.62** | 0.31 | 0.00 | 0.41 | 0.35 |
| Tropical (A) | 289 | 0.40 | 0.31 | 0.25 | 0.22 | 0.43 | **0.53** | 0.26 | −0.01 | 0.31 | 0.13 |
| Dry (B) | 384 | 0.17 | 0.12 | 0.23 | 0.12 | 0.25 | **0.26** | 0.12 | −0.00 | 0.18 | 0.17 |
| Temperate (C) | 3491 | 0.48 | 0.44 | 0.59 | 0.27 | 0.60 | **0.67** | 0.30 | −0.01 | 0.45 | 0.30 |
| Cold (D) | 4041 | 0.44 | −0.05 | 0.53 | 0.28 | 0.58 | **0.61** | 0.33 | 0.04 | 0.39 | 0.42 |
| Polar (E) | 14 | 0.17 | −2.62 | −0.14 | 0.23 | **0.52** | 0.42 | 0.19 | 0.14 | 0.17 | 0.32 |

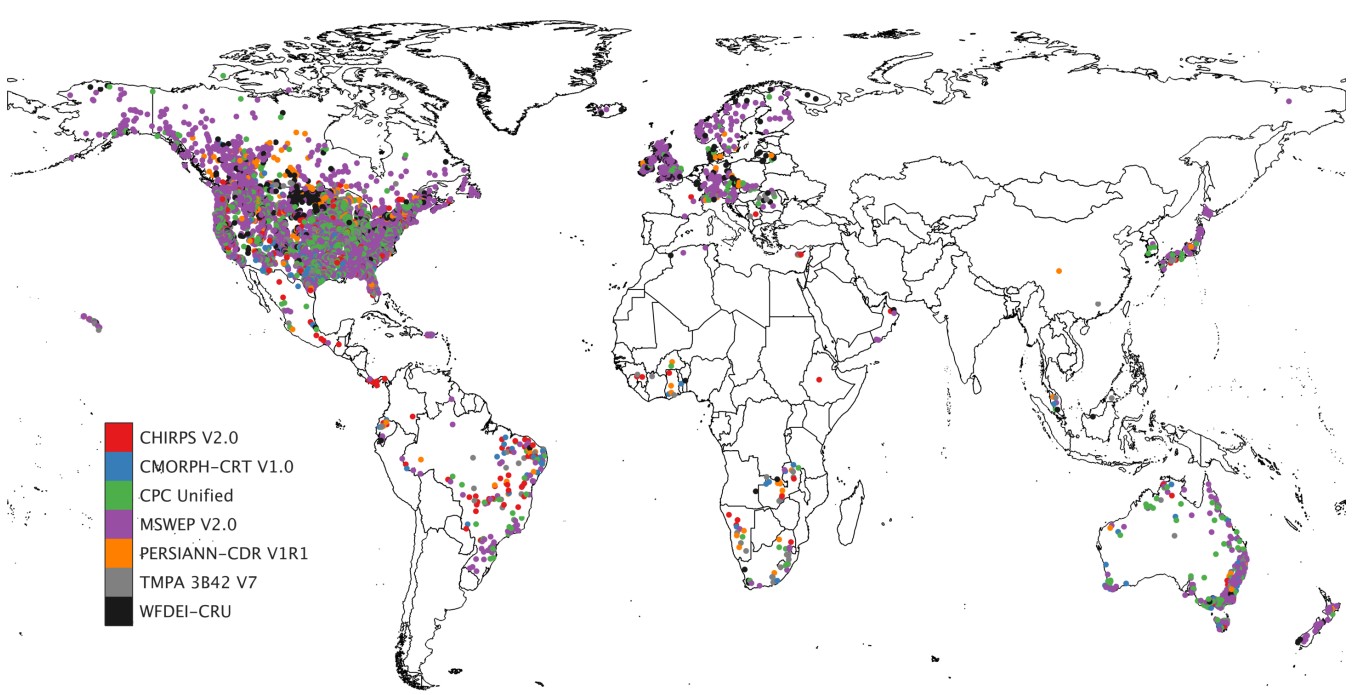

**Figure 3.** For each catchment, the $P$ dataset with the highest calibration NSE. Each data point represents a catchment centroid. Only the seven best performing $P$ datasets (excluding MSWEP V1.2 due to its similarity with V2.0) are considered. Note that CHIRPS V2.0, CMORPH-CRT V1.0, PERSIANN-CDR V1R1, and TMPA 3B42 V7 do not provide data beyond $50°$, $60°$, $60°$, and $50°$ latitude, respectively.





## 4 Conclusions

This study may represent the most comprehensive global-scale $P$ dataset evaluation to date. We evaluated thirteen uncorrected $P$ datasets using $P$ observations from 76 086 gauges, and ten gauge-corrected ones using hydrological modeling for 9053 catchments ($< 50 000$ km$^2$). Our results can be summarized as follows:

1. Among the non-gauge-corrected $P$ datasets, MSWEP-ng V1.2 and V2.0, based on optimal merging of multiple satellite and reanalysis $P$ datasets, provided the best temporal correlations overall. They were followed, in order, by reanalyses, estimates based on microwave remote sensing of rainfall and near-surface soil moisture, and estimates based on thermal remote sensing. MSWEP-ng V2.0 obtained considerably lower mean annual $P$ trend errors than the other datasets. Contrary to expectations, two of the three reanalyses (ERA-Interim and JRA-55) provided, on average, more reliable mean annual $P$ trends than the satellite datasets.

2. Among the uncorrected $P$ datasets, CHIRP V2.0 and MSWEP-ng V1.2 and V2.0 yielded the most accurate long-term $P$ means, primarily due to the use of high-resolution gauge-based climatic datasets to determine their long-term mean. The reanalyses also provided reasonably accurate long-term means. The uncertainty in long-term means among the datasets was generally greatest in topographically complex and arid regions. In terms of the annual number of dry days, MSWEP-ng V2.0 exhibited markedly better performance than the other datasets, due to the use of a cumulative distribution function correction after data merging. The satellite datasets also performed quite well in this respect, while CHIRP V2.0, the reanalyses, MSWEP-ng V1.2, and the soil moisture remote sensing-based SM2RAIN-ASCAT consistently underestimated the number of dry days. The satellite-based datasets generally exhibited difficulties in detecting $P$ signals at high latitudes.

3. Among the gauge-corrected $P$ datasets, the datasets directly incorporating daily gauge data (CPC Unified and the MSWEP versions) outperformed those directly incorporating temporally coarser gauge data. These in turn outperformed the datasets that only indirectly incorporated gauge data. This highlights the benefit of explicit and careful incorporation of daily gauge data. The good performance of the fully gauge-based CPC Unified is unlikely to generalize to sparsely or ungauged regions. In general, the performance was best in temperate regions, due to the presence of dense monitoring networks, and worst in arid regions, due to the convective rainfall and the highly non-linear rainfall-runoff response.

*Acknowledgements.* We gratefully acknowledge the $P$ dataset developers for producing and making available their datasets. The Water Center for Arid and Semi-Arid Zones in Latin America and the Caribbean (CAZALAC) and the Centro de Ciencia del Clima y la Resiliencia (CR)2 (FONDAP 15110009) are thanked for sharing the Mexican and Chilean gauge data, respectively. We also acknowledge the gauge data providers in the Latin American Climate Assessment & Dataset (LACA&D) project: IDEAM (Colombia), INAMEH (Venezuela), INAMHI (Ecuador), SENAMHI (Peru), SENAMHI (Bolivia), and DMC (Chile). We further wish to thank Ali Alijanian, Koen Verbist, and Piyush Jain for providing additional gauge data. The Global Runoff Data Centre (GRDC) and the United States Geological Survey (USGS) are gratefully acknowledged for providing the majority of the observed $Q$ data. We thank Mauricio Zambrano Bigiarini, Pete Peterson, Hamed



Ashouri, and Tomoo Ushio for comments and suggestions. The work was supported through IPA support for the first author from the U.S. Army Corps of Engineers' International Center for Integrated Water Resources Management (ICIWaRM), under the auspices of UNESCO, to further develop a Latin America and Caribbean Drought Monitor; and from NOAA Grant NA14OAR4310219 (Development of a Global Flood and Drought Catalogue for the 20th and 21st Centuries).





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
