# Peer review of "Global-scale evaluation of 22 precipitation datasets using gauge observations and hydrological modeling"

_Hydrology and Earth System Sciences, 2017_

## Referee Comment (RC1) · Anonymous Referee #1 · 19 Sep 2017

Comments: I found this article to be interesting and well-written. It provides a concise review of a number of global scale precipitation datasets and their performance relative to each other. I believe this is work is of broad interest. My primary issue with the manuscript lies in the conclusion section. I would like to see an additional paragraph where the authors synthesize the conclusions presented as points 1, 2, and 3 into specific recommendations about which dataset offers the best performance under various conditions. The authors may wish to discuss performance by climatic classification, geographic region, temporal resolution, etc. Much of these results are found elsewhere in the manuscript, but it would be useful to the reader to have those data in one location. Further, this will provide the opportunity to wrap up the conclusion section which

seems to abruptly end in its current form.

---

## Referee Comment (RC2) · Anonymous Referee #2 · 12 Oct 2017

General comments:

Beck et al. evaluated the performance of 23 precipitation datasets using gauge observations and a HBV hydrological model. The paper fits very well within the stated scope of journal and I read the paper with great interest. The authors deserve considerable credit in taking this extensive study and producing a concise manuscript.

However, I would like to address some suggestions:

- I believe that this manuscript will become more useful if the authors can give further breakdown and more deep analyses for their result in Table 2, e.g. by classifying it to several continents/regions or several climate regions (e.g. as done for Table 3).

[Figure]

- The authors used only NSE for their evaluation using HBV model calibration (while they used several metrics for evaluating P datasets to gauge observations). I am just wondering why the authors selected NSE (among many other measures) for their calibration exercise.

One of the concerns of using NSE, which is a normalization form of the mean squared error (MSE), is its reputation that emphasizes high flows (Legates and McCabe, 1999; Krause et al., 2005). The disadvantage of NSE is the fact that errors between observed and modeled values are calculates as squared values. Consequently, NSE is overly sensitive to large values in time series (whereas lower values are less important). Gupta et al. (1999) mentioned other weaknesses of NSE. One of them is the fact the bias component in NSE is normalized by the standard deviation (i.e. variability) in the observed flows. This means that the bias in time series with high flow variability tends to have little influence in the optimization of NSE, possibly leading to simulations having large volume balance errors. There are many other studies (see e.g. Schaefli and Gupta, 2007) discussing potential problems of using NSE and even Beck et al. (2016) acknowledged NSE as a week metric.

Note that by providing this comment, I am not necessarily suggesting that the authors have to repeat their calibration exercise with different objective functions (which may be very computationally expensive). Rather, I would like to recommend that the authors should validate their existing calibrated runs (already chosen based on their NSE optimization) by calculating some other metrics, e.g. KGE, MAE (mean absolute error), or log NSE. I believe that such validation will make this study more convincing. One can even speculate that an evaluation using log NSE, which emphasize low flow periods, may confirm one of their findings: the superiority of the MSWEP datasets v2.0, which has the best performance in terms of annual dry day error (Table 2).

Details / specific comments:

Section 2.1: I suggest that the authors add brief description for each P dataset. I

believe that this will help readers and improve the quality of the manuscript. Such an explanation can be relatively short as there are similar datasets that can be grouped together, e.g. CHRP and CHRPS, CMORPH and CMORPH-CRT, and all MSWEP datasets.

Table 1: - Please also clarify what the difference between Land and Global. Does the latter include ocean? - Please also explain in the text about the subscript –ng for MSWEP.

Section 2.2: - Page 5, lines 5-7. Here you decided to use MAE, instead of RMSE. I am just wondering why you used NSE, a similar criteria as RMSE, for your performance evaluation using hydrological modelling (Section 2.3)?

Section 2.3: - Why did you use NSE? - Why did you use exclude large catchments (> 50,000 km2)? - If there are several stations along a river (e.g. Meuse), did you use only the most downstream one? Please clarify.

Table 2: Further breakdowns into several continents or climate regions will be useful.

Section 3.1: - Page 7, lines 1-2: MSWEP V2.0 obtained substantially lower mean annual P trend errors than the other P datasets (Table 2 and Supplementary information Figure S5). Please remove "substantially" as the range of these errors is relatively small (as also stated in lines 11-12). - Related to annual P trend errors, I am also wondering what the results will be if longer time series (e.g. starting from 1981) are used.

Section 3.2: I believe that it is more useful to classify and analyze the performances over different climate regions (or continents).

Page 7, line 29: I am curious with the paper Beck et al. (2017a), which is still in preparation.

Section 3.3: - Page 8, lines 23-26. This just shows the superiority of MSWEP datasets. Can you please confirm this superiority for other metrics, e.g. KGE and log NSE. - I am

also wondering what the results will be if longer time series (e.g. starting from 1981) are used. Can you please discuss this?

Table 3: Please improve the caption. What do the letters A, B, C, D and E stand for?

References:

Beck, H. E., A. I. J. M. van Dijk, A. de Roo, D. G. Miralles, T. R. McVicar, J. Schellekens, and L. A. Bruijnzeel, 2016, Global-scale regionalization of hydrologic model parameters, Water Resour. Res.,52, 3599–3622, doi:10.1002/2015WR018247.

Gupta, H. V., H. Kling, K. K. Yilmaz, and G. F. Martinez, 2009, Decomposition of the mean squared error and NSE performance criteria: Implications for improving hydrological modelling, J. Hydrol., 377(1–2), 80–91.

Krause et al., 2005, Comparison of different efficiency criteria for hydrological model assessment, Adv. Geosci., 5(89), 89–97.

Legates, D.R., McCabe, G.J., 1999. Evaluating the use of "goodness-of-fit" measures in hydrologic and hydroclimatic model evaluation. Water Resources Research 35, 233–241.

Schaefli, B., and H. V. Gupta, 2007, Do Nash values have value?, Hydrol. Processes,21(15), 2075–2080

---

## Referee Comment (RC3) · Anonymous Referee #3 · 18 Oct 2017

General comments:

This paper examines 23 near or global precipitation datasets and performs two types of validation. First, validation against gauge data is done for products that do not directly use gauge data in them (no gauge datasets); a calibrated (for each datasets) hydrologic model is compared to streamflow observations around the world for precipitation datasets that incorporate gauge data. This provides independent validation for both types of global precipitation products.

Overall the article is easy to ready and the tables and figures are generally informative. This is a nice contribution to our general knowledge of performance of many widely

used precipitation products. I think this article will be suitable for publication after the authors address my specific comments below.

Major comments:

The discussion should try to answer the "why are the products different, or why do they behave as they do?" question(s) more. Large inter-comparisons are nice to highlight differences, but without answers to "why?" we don't learn much as a field (e.g. the many model inter-comparison studies). I think more effort in discussing this issue throughout will be very beneficial to the final paper. Examples would be the statements on page 6 lines 17-18 regarding GridSatV1.0, and lines 30-31 regarding the reanalyses.

Another concern for me is the reference to the MSWEPv2 paper. It is stated this paper is in prep, not even in review. It is fine to evaluate the new version of a product here, but to have multiple key references to a paper discussing methodological changes, data sources, etc. that is not even in review is not acceptable to me. I understand the desire to publish an evaluation of a product quickly after it is released, but some type of remedy to this needs to be found.

Minor comments:

Page 6, lines 32-33. Is there any comment on the gauge quality in Africa? Could low quality obs be the cause of the poor R3 scores?

Page 7 lines 1-13: Was any quality control done on the trends in the observations? There may be spurious trends in the gauge data.

Page 7, lines 4-5. It should be easy to determine if the assimilated data in the reanalyses changed significantly in the 2000-2016 period, it would be nice to check.

Page 11, lines 10-11. Could the issues in Hawaii be from a missing process in HBV? Can HBV account for deep percolation? Much of the water in Hawaii is lost below the channel network and moves to the ocean in the subsurface.

---

## Author Comment (AC1) · 19 Oct 2017

Dear Dr. Slater,

We hereby provide responses to the reviewer comments for our manuscript entitled "Global-scale evaluation of 23 precipitation datasets using gauge observations and hydrological modeling" in green font below. We have numbered the specific comments for clarity. We would like to sincerely thank you for handling the manuscript in such a prompt and efficient manner.

Sincerely,

Hylke Beck (on behalf of all co-authors)

**Reviewer #1**

C1: I found this article to be interesting and well-written. It provides a concise review of a number of global scale precipitation datasets and their performance relative to each other. I believe this is work is of broad interest. My primary issue with the manuscript lies in the conclusion section. I would like to see an additional paragraph where the authors synthesize the conclusions presented as points 1, 2, and 3 into specific recommendations about which dataset offers the best performance under various conditions. The authors may wish to discuss performance by climatic classification, geographic region, temporal resolution, etc. Much of these results are found elsewhere in the manuscript, but it would be useful to the reader to have those data in one location. Further, this will provide the opportunity to wrap up the conclusion section which seems to abruptly end in its current form.

R1: We thank the reviewer for their compliments and thoughtful suggestion. We agree and have added the following to conclude the Conclusion section: *"So, which precipitation dataset should one use? While this depends on the region under consideration and the specific user needs or application, in most cases MSWEP V2.0 appears to be a good choice: it has a long temporal record (1979-2016), a fully global coverage (including ocean areas), a comparatively high temporal (3-hourly) and spatial (0.1°) resolution, daily gauge corrections, and, as demonstrated in the current study, exhibits comparatively good performance for all performance metrics for all climate types. However, for tropical regions, CHIRPS V2.0 also presents a viable choice, if a daily temporal resolution suffices, and if the peak magnitude underestimation and spurious drizzle are less critical. In regions with dense rain gauge networks, CPC Unified also offers good performance. For some regions, notably Africa, it remains difficult to provide reliable recommendations due to the limited availability and quality of rain gauge and Q data, highlighting the critical importance of maintaining and expanding data collection efforts. However, users who have additional verification data are encouraged to compare the best performing data to ensure that they are adequate for their needs (and share their findings). "*

**Reviewer #2**

General comments:

C2: Beck et al. evaluated the performance of 23 precipitation datasets using gauge observations and a HBV hydrological model. The paper fits very well within the stated scope of journal and I read the paper with great interest. The authors deserve considerable credit in taking this extensive study and producing a concise manuscript.

R2: We want to thank the reviewer for their kind words and constructive suggestions, which helped improve the quality of the paper.

However, I would like to address some suggestions:

C3: - I believe that this manuscript will become more useful if the authors can give further breakdown and more deep analyses for their result in Table 2, e.g. by classifying it to several continents/regions or several climate regions (e.g. as done for Table 3).

R3: We do already discuss performance explicitly for some regions (e.g., *"All datasets performed relatively poorly ($R_{3\ day}$<0.5) in arid and tropical regions ..."*, *"all datasets performed relatively well ($R_{3\ day}$≥0.5) in moist mid-latitude regions ..."*, and *"Africa showed the lowest $R_{3\ day}$ values overall ..."*). Splitting Table 2 into more than the current two groups (<40° latitude and >40° latitude) would make the table too large to fit on a single page. More importantly, it might also overwhelm the reader with values (there are already quite a large number of values) and render the paper substantially less concise. Nevertheless, the actual values underlying the table are presented in the maps (see Figures 1 and 2, and the Supplementary information), providing readers with the possibility of comparing the performance among datasets in any way they see fit. We are also happy to publish the raw data of the comparison so that individuals can make their own assessment.

C4: - The authors used only NSE for their evaluation using HBV model calibration (while they used several metrics for evaluating P datasets to gauge observations). I am just wondering why the authors selected NSE (among many other measures) for their calibration exercise.

R4: We appreciate the comment and added the following text to the manuscript to justify our choice: *"We used the NSE, despite the criticism it has received (e.g., Schaefli et al., 2007, Jain et al., 2008, Criss et al., 2008, Gupta et al., 2009), because: (i) it is highly sensitive to peak flows (Krause et al., 2005), which is desirable for this study given that peak flows are primarily driven by the precipitation forcing whereas low flows are primarily driven by the hydrological model structure and parameters; (ii) besides peak flows, NSE is also sensitive to the long-term bias (Gupta et al., 2009), another important feature of the hydrograph primarily influenced by the precipitation forcing; and (iii) most hydrologists and meteorologists are familiar with the NSE (Moriasi et al., 2007), facilitating the interpretation of the obtained values."*

C5: One of the concerns of using NSE, which is a normalization form of the mean squared error (MSE), is its reputation that emphasizes high flows (Legates and McCabe, 1999; Krause et al., 2005). The disadvantage of NSE is the fact that errors between observed and modeled values are calculated as squared values. Consequently, NSE is overly sensitive to large values in time series (whereas lower values are less important). Gupta et al. (1999) mentioned other weaknesses of NSE. One of them is the fact the bias component in NSE is normalized by the standard deviation (i.e. variability) in the observed flows. This means that the bias in time series with high flow variability tends to have little influence in the optimization of NSE, possibly leading to simulations having large volume balance errors. There are many other studies (see e.g. Schaefli and Gupta, 2007) discussing potential problems of using NSE and even Beck et al. (2016) acknowledged NSE as a weak metric.

R5: We agree with the criticism of the NSE metric and, as mentioned by the reviewer, have even criticized it in our previous papers. However, it is suitable for our purposes in this study. Nevertheless, we now refer to four studies criticizing the NSE (see our preceding response).

C6: Note that by providing this comment, I am not necessarily suggesting that the authors have to repeat their calibration exercise with different objective functions (which may be very computationally expensive). Rather, I would like to recommend that the authors should validate their existing calibrated runs (already chosen based on their NSE optimization) by calculating some other metrics, e.g. KGE, MAE (mean absolute error), or log NSE. I believe that such validation will make this study more convincing. One can even speculate that an evaluation using log NSE, which emphasize low flow periods, may confirm one of their findings: the superiority of the MSWEP datasets v2.0, which has the best performance in terms of annual dry day error (Table 2).

R6: We appreciate the suggestion. However, since the NSE has been used as objective function for the calibration, we do not believe that, without recalibrating each catchment, we will obtain important additional insight by calculating values for other performance metrics. The log NSE, for example, would not provide any additional information, because low flows have not been the focus of the calibration and are thus very poorly constrained. The KGE, which like the NSE is very sensitive to peak flows, would probably yield very similar results.

To properly examine other performance metrics, we would have to recalibrate the hydrological model (which we agree with the reviewer is beyond the scope of the current manuscript). This is due to the simplicity and conceptual basis of the employed hydrological model: it is run in a lumped (non-distributed) fashion, vegetation effects are not explicitly incorporated, routing effects are simulated using a simple post-processing filter, lakes, reservoirs, and wetlands are not explicitly accounted for, etc. However, that is not to say that the model does not serve its purpose, as it is very computationally efficient, enabling us to calibrate thousands of catchments for many precipitation datasets, and flexible, producing satisfactory simulations under a wide variety of climatic and physiographic conditions.

Finally, a validation exercise of our results is unnecessary, given the (unprecedentedly) large number of catchments in all climatic groups (with the exception of the polar group) and the large differences in performance among datasets.

Details / specific comments:

C7: Section 2.1: I suggest that the authors add brief description for each believe that this will help readers and improve the quality of the manuscript. Such an explanation can be relatively short as there are similar datasets that can be grouped together, e.g. CHRP and CHRPS, CMORPH and CMORPH-CRT, and all MSWEP datasets.

R7: Thank you for the useful comment. We added the following text: *"We included seven datasets based exclusively on satellite data (CMORPH V1.0, GSMaP, GridSat V1.0, PERSIANN, PERSIANN-CCS, SM2RAIN-ASCAT, and TMPA 3B42RT V7), three based exclusively on reanalysis data (ERA-Interim, JRA-55, and NCEP-CFSR), and three incorporating both satellite and reanalysis data (CHIRP V2.0, and MSWEP-ng V1.2 and V2.0). Among the gauge-corrected datasets, four combined gauge and satellite data (CMORPH-CRT, GPCP-1DD V1.2, PERSIANN-CDR V1R1, and TMPA 3B42 V7), two combined gauge and reanalysis data (PGF and WFDEI-CRU), while three combined gauge, satellite, and reanalysis data (CHIRPS V2.0, and MSWEP V1.2 and V2.0). We also included a fully gauge-based dataset (CPC Unified)."*

C8: Table 1: - Please also clarify what the difference between Land and Global. Does the latter include ocean? - Please also explain in the text about the subscript –ng for MSWEP.

R8: We have added the following text to the caption: *"In the spatial coverage column, "global" indicates fully global coverage including ocean areas, while "land" indicates that the coverage is limited to the terrestrial surface."*

The "ng" subscript for MSWEP is already defined in Table 1.

C9: Section 2.2: - Page 5, lines 5-7. Here you decided to use MAE, instead of RMSE. I am just wondering why you used NSE, a similar criteria as RMSE, for your performance evaluation using hydrological modelling (Section 2.3)?

R9: We assumed that most readers are familiar with both MAE and RMSE, and thus opted for the slightly better MAE. Conversely, we believe that more readers are familiar with NSE than with KGE (the most obvious alternative) and since NSE is a suitable choice for our purposes, we opted for NSE. As stated in response R4, we now explain in the manuscript our motivation for choosing NSE. That said, KGE is also sensitive to peak flows and does not provide a very different insight, hence for simplicity we preferred to use only NSE.

C10: Section 2.3: - Why did you use NSE? - Why did you use exclude large catchments (>50,000 km2)? - If there are several stations along a river (e.g. Meuse), did you use only the most downstream one? Please clarify.

R10: Large catchments were excluded because of the lumped nature of the hydrological model combined with its daily time step. The following was added: *"We only used catchments <50,000 $km^2$ because applying a daily lumped hydrological model in very large catchments would result in spatial averaging of the forcings over very large areas, confounding the daily runoff generation and water balance calculations."* We did not exclude stations if there were multiple along the same river.

C11: Table 2: Further breakdowns into several continents or climate regions will be useful.

R11: We appreciate the comment. Please see response R3.

C12: Section 3.1: - Page 7, lines 1-2: MSWEP V2.0 obtained substantially lower mean annual P trend errors than the other P datasets (Table 2 and Supplementary information Figure S5). Please remove "substantially" as the range of these errors is relatively small (as also stated in lines 11-12).

R12: Done.

C13: - Related to annual P trend errors, I am also wondering what the results will be if longer time series (e.g. starting from 1981) are Used.

R13: We did not focus on the period prior to the year 2000 since most of the evaluated precipitation datasets do not go back that far. In addition, we feel this is beyond the scope of the current manuscript given the immense efforts necessary to calculate statistics for an extra period.

C14: Section 3.2: I believe that it is more useful to classify and analyze the performances over different climate regions (or continents).

R14: Please see our response identified by R11.

C15: Page 7, line 29: I am curious with the paper Beck et al. (2017a), which is still in preparation.

R15: Beck et al. (2017a) will be submitted shortly.

C16: Section 3.3: - Page 8, lines 23-26. This just shows the superiority of MSWEP datasets. Can you please confirm this superiority for other metrics, e.g. KGE and log NSE. - I am also wondering what the results will be if longer time series (e.g. starting from 1981) are used. Can you please discuss this?

R16: Please see response R6 regarding the use of other performance metrics. We have added the following text discussing the performance prior to the year 2000: *"We expect the dataset performance ranking to be similar for the period prior to the year 2000; however, additional studies are necessary to confirm this."*

C17: Table 3: Please improve the caption. What do the letters A, B, C, D and E stand for?

R17: Thank you for the comment. We have expanded the caption with the following: *"The results are grouped according to the five broadest Koppen-Geiger climate categories, commonly referred to using the letters A-E."*

References:

Beck, H. E., A. I. J. M. van Dijk, A. de Roo, D. G. Miralles, T. R. McVicar, J. Schellekens, and L. A. Bruijnzeel, 2016, Global-scale regionalization of hydrologic model parameters, Water Resour. Res.,52, 3599–3622, doi:10.1002/2015WR018247.

Gupta, H. V., H. Kling, K. K. Yilmaz, and G. F. Martinez, 2009, Decomposition of the mean squared error and NSE performance criteria: Implications for improving hydrological modelling, J. Hydrol., 377(1–2), 80–91.

Krause et al., 2005, Comparison of different efficiency criteria for hydrological model assessment, Adv. Geosci., 5(89), 89–97.

Legates, D.R., McCabe, G.J., 1999. Evaluating the use of "goodness-of-fit" measures in hydrologic and hydroclimatic model evaluation. Water Resources Research 35, 233–241.

Schaefli, B., and H. V. Gupta, 2007, Do Nash values have value?, Hydrol. Processes,21(15), 2075–2080

**Reviewer #3**

General comments:

C20: This paper examines 23 near or global precipitation datasets and performs two types of validation. First, validation against gauge data is done for products that do not directly use gauge data in them (no gauge datasets); a calibrated (for each datasets) hydrologic model is compared to streamflow observations around the world for precipitation datasets that incorporate gauge data. This provides independent validation for both types of global precipitation products.

Overall the article is easy to ready and the tables and figures are generally informative. This is a nice contribution to our general knowledge of performance used precipitation products. I think this article will be suitable for publication after the authors address my specific comments below.

R20: We sincerely thank the reviewer for their constructive criticisms and valuable comments, which helped improve the quality of the manuscript.

Major comments:

C21: The discussion should try to answer the "why are the products different, or why do they behave as they do?" question(s) more. Large inter-comparisons are nice to highlight differences, but without answers to "why?" we don't learn much as a field (e.g. the many model inter-comparison studies). I think more effort in discussing this issue throughout will be very beneficial to the final paper. Examples would be the statements on page 6 lines 17-18 regarding GridSatV1.0, and lines 30-31 regarding the reanalyses.

R21: We fully agree that it is important to explain why the different datasets perform differently and have therefore made several changes.

Regarding the performance of GridSat V1.0, the following was added: *"GridSat V1.0 P estimates have been derived by cumulative distribution function (CDF) matching the entire period of infrared data to a reference P distribution. Better results might be obtained by CDF matching on a monthly or seasonal climatological basis, to account for intra-annual variability in the infrared-P relationship."*

We have added the following to explain why reanalyses exhibit lower skill levels than the microwave- and infrared-based satellite datasets in the tropics (and vice versa in colder regions): *"The comparatively high skill of the reanalyses in colder regions reflects the ability of atmospheric models to simulate synoptic-scale weather systems (Haiden et al., 2012; Zhu et al., 2014). The comparatively low skill of the reanalyses in the tropics is attributable to deficiencies in the sub-grid convection parameterization schemes (Arakawa, 2004), as well as issues in the land surface parameterization and unrealistic strengthening and northward displacement of the monsoon cycle (Di Giuseppe et al., 2013). Multi-scale modeling frameworks incorporating high-resolution (<4 km), convection-permitting models, which negate the need for sub-grid convection parameterization schemes, provide a promising way forward in this regard (Prein et al., 2015; Clark et al., 2016)."*

Regarding the lower performance of PERSIANN-CCS compared to GridSat V1.0 and PERSIANN we added the following: *"This indicates that a higher spatial resolution does not necessarily lead to more skillful estimates, and that there may be limited additional value to be gained from extracting cloud-patch characteristics."*

C22: Another concern for me is the reference to the MSWEPv2 paper. It is stated this paper is in prep, not even in review. It is fine to evaluate the new version of a product here, but to have multiple key references to a paper discussing methodological changes, data sources, etc. that is not even in review is not acceptable to me. I understand the desire to publish an evaluation of a product quickly after it is released, but some type of remedy to this needs to be found.

R22: The MSWEP V2 technical documentation, detailing the most important changes, is already available via www.gloh2o.org. The paper will be submitted within two weeks. A draft version of the paper is available upon request by emailing the first author.

Minor comments:

C23: Page 6, lines 32-33. Is there any comment on the gauge quality in Africa? Could low quality obs be the cause of the poor R3 scores?

R23: It is inevitable that the quality of observations varies across the globe due to a multiplicity of factors, including for example local climate. Although quality standards are set by organisations such as WMO, we have no evidence that stations in Africa perform differently to other regions of the globe. We suspect the main reason for the low scores in Africa is the convection-dominated rainfall regime. We have added the following explanation: *"Africa showed the lowest $R_{3day}$ values overall, probably due to the high prevalence of convective rain events over most of the continent (Cecil et al., 2014)."*

C24: Page 7 lines 1-13: Was any quality control done on the trends in the observations? There may be spurious trends in the gauge data.

R24: Thanks for the comment. We did not conduct any quality control specifically focusing on trends because in most cases it is very difficult to distinguish artificial trends from natural trends. It could probably be done by visual inspection of the records, but this is not feasible given the large number of gauges used in our analysis. That said, the trend errors are likely to be of a random (rather than systematic) nature and as such unlikely to affect the performance ranking of the datasets.

C25: Page 7, lines 4-5. It should be easy to determine if the assimilated data in the reanalyses changed significantly in the 2000-2016 period, it would be nice to check.

R25: Agreed; we have changed the text and added three references: *"our evaluation covers a relatively short period (2000-2016) during which the assimilated observations have not changed considerably (Saha et al., 2010; Dee et al., 2011; Kobayashi et al., 2015)."*

C26: Page 11, lines 10-11. Could the issues in Hawaii be from a missing process in HBV? Can HBV account for deep percolation? Much of the water in Hawaii is lost below the channel network and moves to the ocean in the subsurface.

R26: Thank you for the suggestion; we agree that submarine groundwater discharge may be an additional reason for the poor performance. We have changed the text as follows: *"Low calibration scores were also found in […] Hawaii, we suspect due to (i) flow overestimations caused by erroneous rating curves, as visual inspection of the records revealed the presence of drift errors, and (ii) submarine groundwater discharge (Garrison et al., 2003), which is not explicitly accounted for by HBV."*